# TORSIONAL DIFFUSION FOR MOLECULAR CONFORMER GENERATION

**Bowen Jing,**[*] **Gabriele Corso,**[*] **Regina Barzilay, Tommi Jaakkola**
{bjing, gcorso}@mit.edu, {regina, tommi}@csail.mit.edu
Massachusetts Institute of Technology

## ABSTRACT

Diffusion-based generative models generate samples by mapping noise to data via the reversal of a diffusion process that typically consists of independent Gaussian noise in every data coordinate. This diffusion process is, however, not well suited to the fundamental task of molecular conformer generation where the degrees of freedom differentiating conformers lie mostly in torsion angles. We, therefore, propose Torsional Diffusion that generates conformers by leveraging the definition of a diffusion process over the space $\mathbb{T}^m$, a high dimensional torus representing torsion angles, and a $SE(3)$ equivariant model capable of accurately predicting the score over this process. Empirically, we demonstrate that our model outperforms state-of-the-art methods in terms of both diversity and accuracy of generated conformers, reducing the mean minimum RMSD by respectively 32% and 17%. When compared to Gaussian diffusion models, torsional diffusion enables significantly more accurate generation while performing two orders of magnitude fewer inference time-steps.

## 1 INTRODUCTION

Molecules are defined by their molecular graph, i.e., a set of atoms and the covalent bonds between them. However, it is the set of structures that the graph realizes when embedded in 3D space, called *conformers*, that determine many of its properties. Molecular conformer generation—predicting an ensemble or distribution over 3D conformers for a given molecular graph—is, therefore, a fundamental problem in computational chemistry. Existing approaches consist of methods that sample from the underlying potential energy surface, which are accurate but slow; or approaches leveraging chemical heuristics, which are fast but less accurate.

Deep generative models have been explored for molecular conformer generation in the hopes of combining high accuracy with fast sampling. GeoMol (Ganea et al., 2021) recently demonstrated competitive performance with a message-passing neural network and a custom parameterization and assembly procedure. Diffusion generative models (Ho et al., 2020; Song et al., 2021), which learn to reverse a stochastic process transforming the data distribution into noise, have also shown promise on this task (Shi et al., 2021; Luo et al., 2021; Xu et al., 2021b).

An advantage of diffusion models is that they predict incremental corrections to samples, rather than map latent space elements to samples, which can present challenges due to the geometric symmetries of the target domain. Consequently, mature approaches for predicting geometric quantities (i.e., tensors) over point clouds, such as Tensor Field Networks (TFN) (Thomas et al., 2018) and E(3)NN (Geiger et al., 2020) can be easily used to construct a diffusion generative model. GeoDiff (Xu et al., 2021b) demonstrated that such an approach is superior to diffusion modeling over distance matrices. However, a key drawback to the direct application of diffusion models to point clouds is that the forward diffusion consists of independent Gaussian random noise in each data coordinate. Sampling thus consists of denoising a point cloud where atoms are in random initial positions irrespective of the molecular graph. To obtain competitive results, GeoDiff requires a very large number of ($T = 5000$) such denoising steps, followed by further physics-based optimizations.

---

[*]Equal contribution

We argue that this approach is ill-suited for molecular conformer generation, where bond lengths and angles can be determined very quickly and relatively accurately from the graph alone, and the difference between possible conformers lies largely in the torsion angles (Axelrod & Gomez-Bombarelli, 2020). Instead, it is much simpler and more natural to consider each conformer to be parameterized by torsion angles, and learn to reverse a diffusion that occurs only over these *torsion angle* coordinates. This has the effect of significantly reducing the dimensionality of the sample space; the molecules in GEOM-DRUGS, a common conformer generation dataset (Axelrod & Gomez-Bombarelli, 2020), have, on average, $n = 46.2$ atoms, corresponding to a $3n$-dimensional Euclidean space, but only $m = 8.65$ torsion angles of rotatable bonds.

However, $m$ torsion angle coordinates define not a Euclidean space, but rather an $m$-dimensional torus $\mathbb{T}^m$. Thus, we first formulate the forward diffusion, score-matching, reverse diffusion and denoising procedures over the torus $\mathbb{T}^m$. The theoretical extension of diffusion modeling to non-Euclidean manifolds was developed very recently by De Bortoli et al. (2022). We build upon their work to present, to the best of our knowledge, the first extension of diffusion models to a real-world non-Euclidean domain.

Learning a neural network score model over the input space $\mathbb{T}^m$ presents its own challenges. The dimensionality of this space varies between molecular graphs, and all the information about the molecular graph would have to be made available to the score model. Additionally, there is no canonical way to define the torsion angle coordinate about each bond. To circumvent these difficulties, we instead formulate a *torsion update* about a particular bond as a *geometric* (i.e., $SE(3)$-equivariant) property of a 3D point cloud, and use 3D-equivariant networks to directly predict these properties from a point cloud $\mathbb{R}^{3n}$ representation of the conformer.

By combining diffusion over a torus with a novel equivariant score model over point clouds, we achieve state-of-the-art results on the standard GEOM-DRUGS dataset (Axelrod & Gomez-Bombarelli, 2020). Moreover, we can generate samples with as few as 10 denoising steps—nearly 3 orders of magnitude fewer than the Euclidean diffusion approach employed by GeoDiff.

## 2 METHOD

**Toroidal diffusion**   In Euclidean diffusion models, the data distribution $\mathbf{x}(0) \in \mathbb{R}^d$ is the initial distribution for a diffusion process $d\mathbf{x} = \mathbf{f}(\mathbf{x}, t) \ dt + g(t) \ d\mathbf{w} \quad t \in (0, T)$ which transforms $\mathbf{x}(0)$ into (approximately) a simple Gaussian $\mathbf{x}(T)$. A neural network trained to model the score $\nabla_{\mathbf{x}} \log p_t(\mathbf{x})$ enables sampling from the reverse diffusion,

$$d\mathbf{x} = \mathbf{f}(\mathbf{x}, t) \ dt - g^2(t) \nabla_{\mathbf{x}} \log p(\mathbf{x}, t) \ dt + g(t) \ d\bar{\mathbf{w}} \tag{1}$$

which transforms samples from the simple Gaussian $\mathbf{x}(T)$ into the data distribution $\mathbf{x}(0)$ (Song et al., 2021; Anderson, 1982). A common choice of diffusion process is $\mathbf{f}(\mathbf{x}, t) = 0, g(t) = \sqrt{\frac{d}{dt}\sigma^2(t)}$ where $\sigma^2(t)$ is the variance of the *heat kernel* $p(\mathbf{x}(t) \mid \mathbf{x}(0))$. Access to this heat kernel is sufficient to train a score model $s_\theta(\mathbf{x}, t)$ via denoising score matching; and for sampling the reverse diffusion via the Euler-Maruyama solver. We refer to Song et al. (2021) for further details.

De Bortoli et al. (2022) extended the diffusion model framework to compact Riemannian manifolds, on which the Brownian diffusion converges to a uniform distribution rather than a Gaussian. In particular, for a manifold $M$ embedded in Euclidean space, and knowledge of the score $\nabla_{\mathbf{x}} \log p(\mathbf{x}, t) \in T_{\mathbf{x}} M \quad \forall \mathbf{x} \in M$, the reverse diffusion (equation 1), with Brownian motion on the manifold, remains valid—i.e., its terminal distribution is the original data distribution $\mathbf{x}(0)$.

While modeling and sampling the heat kernel for general manifolds can be complex, for the toroidal diffusion we leverage the fact that $\mathbb{T}^m \cong [-\pi, \pi)^m$ is the quotient space of $\mathbb{R}^m$ with equivalence relations $(x_1, \ldots x_m) \sim (x_1 + 2\pi, \ldots, x_m) \ldots \sim (x_1, \ldots x_m + 2\pi)$. Hence, the heat kernel on $\mathbb{T}^m$ is a simple wrapping of the heat kernel on $\mathbb{R}^m$; that is, for any $\mathbf{x}, \mathbf{x}' \in [-\pi, \pi)^m$, we have

$$p(\mathbf{x}' \mid \mathbf{x}; \sigma^2(t)) \propto \sum_{\mathbf{d} \in \mathbb{Z}^m}^{\infty} \exp\left(-\frac{||\mathbf{x} - \mathbf{x}' + 2\pi\mathbf{d}||^2}{2\sigma^2(t)}\right) \tag{2}$$

We can therefore compute the scores of the heat kernel via a numerical approximation, enabling the training of a score model via denoising score matching. The formalism for sampling from the

reverse diffusion via the Euler-Maruyama sampler is generalized to Riemannian manifolds in terms of a geodesic random walk (De Bortoli et al., 2022), which is also simplified in the case of the torus as the wrapping of the random walk on $\mathbb{R}^m$.

**Conformer diffusion** A molecule is a graph $G = (\mathcal{V}, \mathcal{E}) \in \mathcal{G}$ with atoms $v \in \mathcal{V}$ and bonds $e \in \mathcal{E}$, with $n = |\mathcal{V}|$. A conformer $c \in \mathcal{C}_G$ of a molecule is a vector in $\mathbb{R}^{3n}$ defined (up to global rototranslation) by $G$ along with chirality tags $\mathbf{z} \in \{-1, 1\}^k$ where $k$ is the number of chiral centers, a set $L$ of bond lengths and angles, and a set of *torsion angles* $T = (t_1, t_2, \ldots t_m) \in [-\pi, \pi)^m = \mathbb{T}^m$ where $m$ is the number of rotatable bonds. The assembly of 3D coordinates from these *intrinsic coordinates* $L, T$ is thus a function $F_G : \{-1.1\}^k \times \mathcal{L} \times \mathbb{T}^m \mapsto \mathcal{C}_G$ where each $c \in \mathcal{C}_G$ is a set $c = \{g(\mathbf{x}) \mid g \in SE(3)\}$ for some $\mathbf{x} \in \mathbb{R}^{3n}$. Given this parameterization and with $\mathbf{z}, L$ fixed in advance, a diffusion over $\mathbb{T}^m$ maps to a diffusion on $\mathcal{C}_G$.

We now desire a score model which maps from $\mathcal{C}_G$ to the tangent space of $\mathbb{T}^m$, which is isomorphic to $\mathbb{R}^m$. We therefore use an $SE(3)$-equivariant score model (Geiger et al., 2020) conditioned on the input graph $s_G : \mathbb{R}^{3n} \mapsto \mathbb{R}^m$, which can be viewed as a function over $\mathcal{C}_G$ since for any $c \in \mathcal{C}_G$ and any $\mathbf{x}, \mathbf{x}' \in c$, we have $s_G(\mathbf{x}) = s_G(\mathbf{x}')$. An additional symmetry arises from the fact that the underlying physical energy is invariant under *parity inversion*; thus our learned density should respect $p_G(-c) = p_G(c)$ where $-c = \{-\mathbf{x} \mid \mathbf{x} \in c\}$. The intrinsic coordinates transform under parity inversion as $\mathbf{z} \mapsto -\mathbf{z}, L \mapsto L, T \mapsto -T$, where the latter can be seen from the formula for the torsion angle $\angle(ABC, ABD)$ (Ganea et al., 2021):

$$\angle(ABC, ABD) = \arctan 2 \left( \frac{||x_B - x_A||(x_A - x_C) \cdot [(x_B - x_A) \times (x_D - x_B)]}{[(x_A - x_C) \times (x_B - x_A)] \cdot [(x_B - x_A) \times (x_D - x_B)]} \right) \quad (3)$$

Therefore, parity inversion invariance is respected if the learned density over the torsion angles $\mathbb{T}^n$ respects $p_G(T; L, \mathbf{z}) = p_G(-T; L, -\mathbf{z})$. Since an invariant density corresponds to an *equivariant* score, we need $s_G(F_G(T, L, \mathbf{z})) = -s_G(F_G(-T, L, -\mathbf{z}))$, which implies $s_G(\mathbf{x}) = -s_G(-\mathbf{x})$. Thus, the score model must be *invariant* under $SE(3)$ but *equivariant* under parity inversion of the input point cloud—i.e., it must output a set of *pseudoscalars*. Note that both the 3D coordinates $\mathbf{x}$ and the score $s_G(\mathbf{x})$ do not depend on the parametrization of each torsion angle $t_i$ (based on the arbitrary choice of neighbors) which, therefore, we can avoid and are invariant to.

**Score model** The architecture we designed to predict is formed by three components: an embedding layer, a series of interaction layers and a torque layer. In the embedding layer, we build a radius graph around each atom on top of the original molecular graph and generate initial embeddings for nodes and edges combining chemical properties, sinusoidal embeddings of $t$ and, for the edges, a radial basis function representation of their length.

The interaction layers are based on TFN convolutional layers. At each layer, for every pair of nodes in the graph, we construct messages using tensor products the current irreducible representation of each node with the second-order spherical harmonics representation of the normalized edge vector and weight them. These messages are represented using irreducible representations up to the 2$^{\text{nd}}$ rotation-order and weighted channel-wise by a scalar function of the current scalar representations of the two nodes and the edge. Finally, these messages are averaged to obtain new irreducible representation for each node.

Finally, in the torque layer, for every rotatable bond, we construct a tensor-valued filter from the tensor product of the spherical harmonics with a $L = 2$ representation of the *bond axis*. This filter is then used to convolve with the representations of every neighbor on a radius graph, and the pseudoscalar products are passed through odd-function (i.e., with $\tanh$ nonlinearity and no bias) dense layers to produce a single prediction.

## 3 EXPERIMENTS

**Inference setup** At inference time, in order to use torsional diffusion on a molecule we first need an estimate of the bond lengths and angles of its conformers. This estimate can be done with high accuracy with rule-based methods. Therefore we use RDKit ETKDG (Riniker & Landrum, 2015) to generate a set of conformations, and then sample uniformly random initial conformations in the toroidal space by randomly changing each of its torsion angles by $U[-\pi, \pi]$, while keeping the same

Table 1: Performance of various methods on the GEOM-Drugs dataset test-set. Note that GeoDiff uses a different set of random splits.

| Model | Cov-R ↑ | | AMR-R ↓ | | Cov-P ↑ | | AMR-P ↓ | |
|---|---|---|---|---|---|---|---|---|
| | Mean | Med | Mean | Med | Mean | Med | Mean | Med |
| RDKit ETKDG | 68.78 | 76.04 | 1.042 | 0.982 | 71.06 | 88.24 | 1.036 | 0.943 |
| OMEGA | 81.64 | 97.25 | 0.851 | 0.771 | 77.18 | **96.15** | 0.951 | 0.854 |
| CGCF | 54.35 | 56.74 | 1.248 | 1.224 | 24.48 | 15.00 | 1.837 | 1.829 |
| GeoMol | 82.43 | 95.10 | 0.862 | 0.837 | 78.52 | 94.4 | 0.933 | 0.856 |
| GeoDiff | 89.13 | 97.88 | 0.863 | 0.853 | 61.47 | 64.55 | 1.171 | 1.123 |
| Torsional Diffusion (ours) | **96.32** | **100** | **0.582** | **0.565** | **84.90** | 94.38 | **0.778** | **0.729** |

bond lengths and angles. Conformers are then sampled from the learned model by performing 20 steps of reverse diffusion with the Euler-Maruyama reverse SDE sampler.

**Training setup**    At training time, if we directly diffuse the ground truth conformers, the model in its diffusion process would have access to the exact bond lengths and angles. Our experiments show that the disparity between training and inference of having exact versus estimated bond lengths and angles causes a distributional shift that hurts the model performance at inference time. We bridge this shift at training time with the following procedure: as a preprocessing step, we substitute the true conformers with conformers generated by matching the ones generated with ETKDG to the true conformer with a torsion angle differential evolution fitting procedure.

**Dataset & evaluation**    We test our method on the GEOM-DRUGS dataset (Axelrod & Gomez-Bombarelli, 2020), which is composed of a total of 304k molecules each with an associated set of conformers obtained using CREST. To provide a fair comparison with previous methods, we follow the filtering, splitting and evaluation metrics from Ganea et al. (2021). We report Average Minimum RMSD (AMR) and Coverage (COV) both for Recall (R) —how many ground truth conformers are correctly predicted— and Precision (P) —how many of the predicted structures are of high quality.

**Baselines**    We compare our performance with a wide variety of existing methods. RDKit ETKDG (Riniker & Landrum, 2015) is the most popular open-source method. OMEGA (Hawkins et al., 2010; Hawkins & Nicholls, 2012) is a rule-based commercial package in continuous development. CGCF (Xu et al., 2021a), GeoMol (Ganea et al., 2021) and GeoDiff (Xu et al., 2021b) are recent machine learning approaches that have achieved competitive or state-of-the-art performances.

**Results & discussion**    The results presented in Table 1 show that our method obtains state-of-the-art results in most precision and recall metrics, reducing the mean AMR by 17% and 32% for precision and recall respectively. This highlights the strength of the presented method and the high diversity obtainable with diffusion models. The advantage provided by the torsional diffusion is evident when comparing it to the other method based on diffusion models, GeoDiff, operating in Euclidean space. GeoDiff requires 5000 inference steps, and therefore 5000 score model evaluations, to obtain the results in Table 1. On the other hand, torsional diffusion is able to significantly outperform it on all metrics with only 20 steps by restricting the diffusion process in the subspace where most of the molecule's flexibility lies.

## 4    CONCLUSION

We presented torsional diffusion, a novel method based on score-based diffusion models, to generate molecular conformers. We define a diffusion process of the high dimensional torus representing the space of possible torsion angles and the associated score-matching and reverse diffusion. Then, we present a novel $SE(3)$-equivariant model to predict the scores in the torsion angle distributions and a preprocessing technique to bridge the inference distributional shift. Empirically, we obtain state-of-the-art performances on precision and recall metrics. Moreover, unlike previous diffusion-based techniques on Euclidean spaces, we are able to generate conformers with almost two orders of magnitude fewer time-steps.

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
