# OpenReview forum: "Torsional Diffusion for Molecular Conformer Generation"
_ICLR.cc/2022/Workshop/DGM4HSD — ICLR 2022 DGM4HSD workshop Oral_

### Official Review · Reviewer_sU7z · 2022-03-20
**Recommendation to accept**

**Rating:** 8
**Confidence:** 4

**Review:**

#####Summary#####

This work provides a diffusion model based on the space of possible torsion angles for molecular conformer generation. The idea is intuitively sound and straightforward. It can significantly reduce the sampling complexity. The experiments can demonstrate the performance quite well.

#####Pros#####

(1) The proposed idea about considering the degree of freedom on torsion angles are quite intuitive, straightforward, and technically reasonable for generating conformers. It significantly reduces the number of degrees of freedom as in previous works that consider all atoms’ 3D coordinates.

(2) The experimental results are convincing to me. It can show the obvious improvements over prior works.


#####Cons/Suggestions#####

(1) As mentioned in the paper, RDkit ETKDG is required to firstly determine the bond lengths and angles. Hence, this step should count for the total complexity of the proposed method. It is highly suggested to consider this part when comparing with baselines in term of complexity. This would provide a better illustration of the complexity of this method.

(2) The method part is good from a high-level perspective. However, more details should be clarified to make it clearer.

---

### Official Review · Reviewer_DNtf · 2022-03-22
**An innovative diffusion-based generative approach for conformers**

**Rating:** 9
**Confidence:** 2

**Review:**

This paper proposes a diffusion-based generative approach for conformers. In contrast to previous approaches, the authors propose to define the diffusion over a torus to represent torsion angles for the generation of conformers -- which provides a more natural parameterization of conformers. They compare their method against previous methods demonstrating promising performance.

Overall, the paper is well-written and the method of defining diffusion process on high-dimensional torus to represent the space of torsion angles of conformers seems novel. The method is well described and the experiments are clear -- to the degree that can be fit on a short 4-page workshop paper.  I am not familiar with the field and so cannot comment on the breadth of the methods that were compared and whether the GEOM-DRUGS dataset suffices to justify generalizable claims.

My only (minor) concern arises from the claims of SOTA performance on all diversity metrics is a bit of a stretch. A more balanced assessment in the conclusions would be more realistic. Perhaps even stating the limitations of this study and what could be further optimized or tweaked for further improvement in future iterations.

---

### Decision · Program_Chairs · 2022-03-27

Accept (Oral)